



# Seismic Hazard of L'Aquila downtown (central Italy): new insights for 3D geological model based on high-resolution seismic reflection profile and borehole stratigraphy

5   Marco Tallini[1], Marco Spadi[1], Domenico Cosentino[2], Marco Nocentini[3], Luca Macerola[1], Giuseppe Cavuoto[4], Vincenzo Di Fiore[4]

1 Dipartimento di Ingegneria Civile, Edile-Architettura, Ambientale, Università degli Studi dell'Aquila, Via Giovanni Gronchi, 18, 67100 L'Aquila, Italy

2 Dipartimento di Scienze, Università degli Studi Roma Tre, Largo San Leonardo Murialdo, 1, 00146 Roma, Italy

3 Istituto di Geologia Ambientale e Geoingegneria - CNR, Via Salaria km 29.300, 00015 Montelibretti, Roma, Italy

4 Istituto per l'Ambiente Marino Costiero–Consiglio Nazionale delle Ricerche, Calata Porta di Massa, 80133 Napoli, Italy

*Correspondence to*: Marco Tallini (email: marco.tallini@univaq.it)

**Abstract.** On April 6, 2009, a Mw 6.1 earthquake struck the Plio-Quaternary intermontane L'Aquila Basin in central Italy, causing strong damages in L'Aquila historical downtown and surroundings, which were affected by notable site effects. Previous works have suggested that different site effects may be related to the complex subsurface geologic architecture, given

by the variability of thickness and lithology of L'Aquila Basin clastic deposits, on which the city was built. To improve the 3D geological model of L'Aquila downtown for seismic site response evaluation and to estimate the Seismic Hazard of possible buried active normal faults, a multitask project has been carried out consisting mainly of the integration of subsurface dataset, including geological and geophysical surveys. Data have been interpreted with the aim to conceive and build a detailed model for the Plio-Quaternary cover of the continental basin and the buried morphology of the Meso-Cenozoic bedrock. We report

the results concerning the interpretation of a 1 km-long high-resolution seismic reflection profile and refraction tomography integrated with the stratigraphy from deep and shallow boreholes. The results allowed us to reconstruct the Plio-Quaternary succession below L'Aquila downtown. The Plio-Quaternary depocentre corresponds to a minor NNW-SSE graben, which is developed within the main regional graben that borders L'Aquila Basin. Finally, data interpretation allowed to reconstruct the Plio-Quaternary tectono-stratigraphic evolution of the basin, to evidence the recent activity of several faults, and to define the

subsoil geological model of the study area. All these data, which are functional to define the seismic site effects and to detect the activity of faults, are useful to mitigate the Seismic Hazard of cultural heritage cities of central Italy, such as the case study of L'Aquila downtown.





## 1 Introduction

Generally, to mitigate the Seismic Hazard of cultural heritage cities of central Italy, which are mainly placed in Plio-Quaternary intermontane basins characterised by high seismicity, as demonstrated by the recent earthquakes (i.e. Mw 6.1 L'Aquila event of April 6, 2009 and Mw 6.0 Amatrice event of August 24, 2016: Gruppo di Lavoro MS–AQ, 2010; Rossi et al., 2019), the

knowledge of the 3D geological model is primary for the evaluation of the seismic local effect and the recognition of active faults. Amplification effect related to the seismic wave propagation are mainly due to vertical and lateral changes in the subsurface stratigraphy and/or abrupt variation in topography (Bard and Gariel, 1986). Considering the regional geology of central Italy, the thickness of Plio-Quaternary covers resting above the Meso-Cenozoic bedrock, generally corresponding to the seismic bedrock, must be taken into account in evaluating the amplification effect (Bordoni et al., 2014; Gaudiosi et al.,

2014; Lanzo et al., 2011; Martelli et al., 2012). In the densely populated intermontane basins of the central Apennines (e.g. Tiberino, Rieti, Leonessa, L'Aquila, Fucino, Sulmona, etc.), which are tectonically active areas bounded by seismogenic normal faults, the thickness of the Plio-Quaternary covers is mainly related to the geometry of the extensional fault systems and their evolution in space and time (Fig. 1). High-resolution dataset on surface and subsurface data, in terms of both geology (i.e. field work on both stratigraphy and tectonics, biostratigraphy, geochronology, boreholes stratigraphy, etc.) and geophysics

(i.e. high-resolution seismic reflection profiles, high-resolution seismic refraction tomography, electrical resistivity tomography, gravimetry, etc.) are needed to create a reliable 3D model, especially in urban area (Di Giulio et al., 2014).

Studies on site effects in urban areas have been greatly increased after the April 6, 2009 Mw 6.1 earthquake that struck L'Aquila Basin, causing strong damages in the L'Aquila downtown (i.e. the cultural heritage urban area encircled inside the medieval walls; in the following AD) (Chiarabba et al., 2009; Gruppo di Lavoro MS–AQ, 2010; Moro et al., 2017). Previous works

suggested that different amplification effects, recorded within AD, could be related to the complex subsurface geologic architecture, mainly due to variability in thickness and lithology of the continental deposits filling the Plio-Quaternary graben, on which AD was built (Del Monaco et al., 2013; Durante et al., 2017; Nocentini et al., 2017; Amoroso et al., 2018).

This paper provides new subsurface data consisting in a high-resolution seismic reflection profile, combined with seismic refraction tomography, along a ca. 1000 m-long N-S trace, crossing AD (named in the following Corso section). The Corso

section was interpreted by using the stratigraphy of several deep and shallow boreholes and fine scale geological field data performed in the last years by different authors (Tallini et al., 2012; Cosentino et al., 2017; Nocentini et al., 2017; 2018). The main result of this work is the reconstruction of the contour lines of the top of Meso-Cenozoic bedrock, which allowed to better define the Plio-Quaternary tectono-stratigraphic evolution of the L'Aquila Basin. In addition, this work contributes to define the seismic site effects (amplification and surface active faulting) at urban scale and to refine the geometry and the activity of

the fault system beneath AD, which are basic to improve the earthquake-resistant restoration on cultural heritage buildings (Chopra Anil, 1995).



## 2 Geological setting

AD is located in the L'Aquila-Scoppito Basin (ASB), which corresponds to the western part of the large L'Aquila Basin, a NW-SE oriented intermontane basin of central Italy. ASB is a E-W–trending half graben, bordered to the north by both the active south-dipping Scoppito-Preturo normal fault and the southwest-dipping Pettino normal fault (Mancini et al., 2012;
Tallini et al., 2012; Storti et al., 2013; Cosentino et al., 2017; Nocentini et al., 2017) (Fig. 1).

In this sector of the central Apennines fold-and-thrust belt, the post-orogenic extensional deformation is documented from Late Pliocene to Present, with the development of normal fault systems, mostly dipping toward S-SW, responsible for the onset and subsequent evolution of the post-orogenic intermontane basins (Giaccio et al., 2012; Porreca et al., 2016; Cosentino et al., 2017, with references therein) (Fig. 1). In the ASB, the Meso-Cenozoic carbonate units (SLB and CRP) and the Upper
Miocene synorogenic terrigenous deposits (SYN) are unconformably overlaid by a thick succession of Plio-Quaternary continental clastic deposits (Fig. 2) (Cosentino et al., 2010; 2017). The facies distribution of these Plio-Quaternary continental deposits has been studied combining field geology and boreholes stratigraphy, allowing to reconstruct the tectono-sedimentary evolution of the ASB (Mancini et al., 2012; Tallini et al., 2012; Cosentino et al., 2017; Nocentini et al., 2017). Despite these recent progresses, the understanding of the geometry and the internal architecture of the Plio-Quaternary sedimentary filling
beneath AD is still incomplete.

The continental sedimentation within the L'Aquila Basin started in the late Piacenzian-Gelasian with the deposition of the San Demetrio-Colle Cantaro Synthem (Fig. 2) (Spadi et al., 2016; Cosentino et al., 2017; Nocentini et al., 2017; 2018;). In the ASB, this synthem is represented by the presence of the Colle Cantaro Fm. (CCF), which consists of heterometric breccias, in abundant clayey-silty matrix, deposited in slope and debris flow environment (Nocentini, 2016; Nocentini et al., 2017).
Unfortunately, CCF never crops out in the proximity of AD. In the field, CCF is differently tilted due to the activity of the main faults and to the deposition in different environments (Centamore and Dramis, 2010). The Madonna della Strada Fm. (MDS) unconformably overlays both CCF and the Meso-Cenozoic bedrock. MDS consists of silt, sand beds, and clayey silts, containing lignite intercalations (Mancini et al., 2012; Storti et al., 2013; Cosentino et al., 2017; Nocentini et al., 2017). In the lower and middle part of MDS, coarse- to medium-grained, well-rounded sandy gravel beds are present, showing variable
thickness and channelized geometries with planar and through crossbedding. The sedimentary structures of MDS clearly point to a deposition in a meandering fluvial system, which developed within the ASB with a wide floodplain and swampy areas during the Calabrian (Early Pleistocene) (Cosentino et al., 2017). Along the border of the MDS alluvial plain, coarse-grained slope deposits sedimented, interfingering the MDS pelite, called San Marco Fm. (SMF) (Nocentini et al., 2017). The Meso-Cenozoic bedrock and the previously described continental units can be covered by the Fosso di Genzano Synthem (FGS).
FGS is composed of coarse- to medium-grained clast-supported gravel beds, with general horizontal bedding and fining-upward gradation with yellowish sands or silty layers. These coarse-grained deposits belong to a braided or wandering fluvial system with lateral alluvial fans (Nocentini et al., 2017). Based on tephra layers (Gaeta et al., 2010) and large mammal remains (Palombo et al., 2010), FGS can be referred to the lower part of the Middle Pleistocene. The main part of AD was built over



the Colle Macchione-L'Aquila Synthem (CMA), which overlies either the older synthems or the Meso-Cenozoic bedrock through a highly irregular and erosive basal surface. CMA mainly consists of highly heterometric breccias with angular carbonate clasts surrounded by whitish calcareous silty matrix. The calcareous breccias of CMA include intercalations of both carbonate silt deposits (eastern margin of AD) and siliciclastic medium- and coarse-grained deposits (southern margin of AD).

The sedimentological characteristics of these carbonate breccias point to debris flows and rock avalanches derived from the mountain slopes surrounding the ASB (Centamore and Dramis, 2010; Esposito et al., 2014; Cosentino et al., 2017). The age of CMA is possibly referred to cold phases of late Middle Pleistocene (Cosentino et al., 2017; Nocentini et al., 2017). The top of L'Aquila hill, on which AD is placed, is capped by Collemaggio Synthem (COM), which mainly consists of reddish to dark brown clayey silt sediments with rare subangular calcareous clasts. COM is interpreted as epikarst fill and reworked paleosols,

probably formed under interglacial conditions, which suggests a correlation with the Eemian interglacial stage (MIS 5e, base of the Late Pleistocene, ~125 ka) (Magaldi and Tallini, 2000; Nocentini et al., 2017). Finally, the younger continental units are alluvial, colluvial, slope, and anthropic deposits of Late Pleistocene and Holocene, which have been grouped in the same supersynthem (ALL).

## 3 Methods and data acquisition

For the reconstruction of the subsurface geology of AD we used both the stratigraphy of several deep and shallow boreholes (depth 30-440 m b.g.l.) (Fig. 3) and geophysical data (seismic reflection profile, named Corso section, and seismic refraction tomography) acquired just after the main shock that on April 6, 2009 struck AD. These subsurface data have been integrated with outcrop data of AD.

Standard vertical profiling of well logs were used for the interpretation of the Corso section. In addition, the structural analysis of the seismic reflection profile followed the typical seismic interpretation methodology in defining reflector terminations (Posamentier and Allen, 1993). According to differences in amplitude, frequency, and continuity of the seismic reflectors, together with both their terminations and lateral distribution, the different seismic facies on the Corso section were recognized.

## Seismic data acquisition

The Corso section has a NNE-SSW strike and a length of 960 meters. A vibratory seismic source, the IVI-MINIVIB shaker truck, was used for the survey. This source vibrates by a 168 kg mass and it can produce harmonic vibrations (sweep) with a maximum peak force of 27600 N. The source move-up was 10 m; at each of the 91 vibration points, three 15 s long, 10-200 Hz sweeps were stacked and recorded by a dense (5 m spacing) 192-channels, 10-Hz vertical geophone array. This multi-fold

wide-angle geometry allowed to collect highly redundant turning waves and deep-penetrating refracted waves, which contain



information on the velocity distribution in depth. The acquisition parameters permitted a maximum common midpoint (CMP) redundancy of 4800% (Tab. 1).

Data were acquired using n. 8, 24-bit, 24-channel GEODE Geometrics seismographs and were recorded by a PC running an applicative code for setting the field acquisition parameters and storing the seismic data in seg-2 format.

Several conventional processing steps were applied to produce CDP stacked sections (Steeples and Miller, 1988; Yilmaz, 2001). The processing sequence is summarized as follows: (i) pre-processing phase: import seg-2 data, cross-correlation, geometry setup, trace editing (Kill bad traces, Top Muting, Bottom muting); (ii) processing phase: amplitude correction, static correction, spiking deconvolution, velocity analysis (semblance), NMO, stack, Kirchhoff time Migration.

Together with seismic reflection processing, first arrival times was picked to perform a non-linear tomography. Tomography

data was used both to extend the seismic imaging at the very near surface (first 30-50 m) since usually this part is not sampled, even by "shallow" seismic reflection techniques and used for the static correction. The seismic refraction tomography was based on a ray-tracing modelling algorithm with a simultaneous iterative reconstruction technique algorithm for inversion (Hayashi and Takahashi, 2001). Then, an initial velocity model was created generating many thin layers composed of quadrangle cells with uniform velocity value. The algorithm calculates the fastest ray connecting nodes defined on the

boundaries of the cells to produce a bidimensional velocity fields that best minimizes the travel-time residuals. A misfit function, consisting of the squared difference between the observed and computed travel times, was calculated. The model was adjusted until the misfit is minimized. The iterations were stopped when the RMS travel time residual (difference between the calculated travel times for the initial model and the observed ones) is less than the average travel time pick error. For refraction data analysis, all first-arrival travel times were accurately hand-picked on the common shot panels. Travel-time

diagrams were created and checked for consistency, following the rules of Ackermann et al. (1986).

## 4 Results

**Boreholes stratigraphy**

Hundreds of boreholes were drilled in the whole ASB to investigate the subsoil deposits following the April 6, 2009 L'Aquila earthquake (Nocentini et al., 2017). We present the stratigraphy of several deep boreholes drilled in the vicinity of the Corso

section, which are useful to constrain the interpretation of the deep subsurface geology of AD as imaged by the seismic reflection profile (Fig. 3). In addition, the stratigraphy of several shallow boreholes drilled along the Corso section were used for constraining the uppermost part of the seismic refraction tomography of AD. Considering the described stratigraphy, a comprehensive review of the available well logs has been carried out to define the lithology and the geometry of the basin fill, as well as to reconstruct the depth variation of the bedrock through the ASB (Nocentini, 2016).

Well logs analysis showed rapid changes in the bedrock depth, resulting in the occurrence of subsurface troughs and ridges within the basin, as well as rapid changes of thickness and facies of the continental deposits, which reflect the extensional tectonic deformation starting from ca. 3 Ma (Cosentino et al., 2017). Well logs highlighted the presence in all the boreholes,





except for the shallower S1 and S8, of MDS lying above an irregular surface carved both in the Meso-Cenozoic bedrock and the CCF (Fig. 3). Though CCF represents the first post-orogenic infilling of ASB it never crops out in AD and it was not found in the total examined boreholes, but it is probably present in the depocentre of ASB as highlighted by seismic reflection profiles (Cosentino et al., 2017; Nocentini et al., 2017). The examined boreholes showed the existence of a deep depocenter,

with a maximum thickness of MDS from well S10, which drilled 400 m of fine-grained siliciclastic deposits referable to alluvial plain environments (well log database available at DICEAA's Applied Geology laboratory of L'Aquila University). Looking at the borehole stratigraphy, the MDS presents a highly variable thickness and shows latero-vertical changes in lithofacies all over the ASB.

The FGS has been found mainly in the boreholes drilled in the southern sector of AD. In other parts of ASB it was found

discontinuously with thicknesses reducing toward the north (Nocentini et al., 2017). In contrast, the coarse-grained deposits of CMA are always present in the boreholes drilled within AD, even though they show variable thickness between 15 m and 70 m, CMA is not present in S12 and S13 boreholes.

The 140 m-deep S5 borehole (Fig. 4) was drilled at the base of the southern slope of AD. The first 17 meters of the sedimentary cores consist of 2 m of anthropogenic and colluvial deposits and less than 15 m of continental breccias pertaining to CMA

(Nocentini, 2016). Below CMA, yellowish sub-horizontal laminated fine-grained sands and silty sands are present. The occurrence of volcanic minerals within these sandy deposits allow to refer them to FGS. In the S5 borehole, a pedogenetic horizon (oxidized surface) distinguishes the FGS from the fine-grained deposits referable to MDS (Nocentini, 2016). These fine-grained deposits consist of massive to laminated sandy silts and sands that pass to alternations of silts, clayey silts, clays, and fine sands with lignite levels toward the base of the borehole. Within the silty sandy layers of the upper portion of MDS,

structures related to soft-sediment deformation, such as highly deformed lamination showing convolute-like or deformed/broken laminae, likely due to fluid expulsion possibly during seismic wave propagation (seismites), are visible in the sedimentary cores of different boreholes (Fig. 4). In addition, in the S5 borehole, the sediments pertaining to MDS show a relatively high sand content (Fig. 4). These sandy layers present high trough and/or planar cross-bedding, sometimes coupled with ripple cross laminations with an erosive basis and scour-and-fill structures. They usually define lenses of conglomerate

and sand bodies that are often separated by structureless pelite.

**Seismic facies recognition**

Based on the parameters of the seismic reflectors, such as geometry, seismic reflection amplitude, continuity, and qualitative reflection frequency, five seismic facies were identified in the Corso section (Fig. 5). All the distinguished seismic facies can be described following qualitative terms (Tab. 2). The seismic facies interpretation was further checked by comparing them

with the lithological information coming from the borehole stratigraphy. To facilitate correlations at a basin scale, in labelling the different seismic facies we applied the same nomenclature used by Cosentino et al. (2017) in the interpretation of the Pettino I seismic reflection profile.



Several discontinuities in the semi-continuous reflectors are interpreted as faults, the main one is the normal fault located at CDP 98 (240 m) that displaced most of AD formations. It is a SW-dipping plane with a listric shape and is located beneath Piazza Duomo (the main square of AD) and is named Piazza Duomo Fault (PDF).

**Seismic facies S.** Seismic facies S is clearly composed by chaotic seismic facies, with low-amplitude dipping reflective pattern, and discontinuous reflectors. It is present on the deepest portion of Corso section. Low-resolution signal of seismic facies S is probably due to penetration problem of the seismic pulse. The weak diffusion connected with low resolution signal and discontinuity of reflectors points to interpret this seismic facies as the acoustic bedrock of ASB.

**Seismic facies R.** Seismic facies R is typically characterized by high reflectivity and amplitude, with disturbed or semi-continuous reflectors. It characterizes the lower part of the Corso section, showing a maximum thickness of ~200 ms. The faintly continuous reflectors show a divergent internal configuration, typical of wedge-shaped geometries. The boundary between this seismic facies and the seismic facies S is irregular, probably due to the extremely disturbed signal, but appears as an erosional contact. The irregularly-fringed and high-amplitude reflectors of seismic facies R possibly can be referable to clastic coarse-grained deposits, including slope breccias, debris-flow, and clayey-sandy conglomerates.

**Seismic facies L.** Seismic facies L is composed by continuous and parallel reflectors with medium to high amplitude. Along the Corso section, this facies has variable thickness between ~300 ms in the northern part and ~450 ms in the central part. Seismic facies L shows horizontal parallel and semi-parallel reflectors, that are tilted and dragged along the extensional faults affecting the Corso section. The seismic facies L corresponds to more sub-parallel and laterally extensive reflectors (HARP's: High Amplitude Reflection packages, or HAC: High Amplitude Continuous) (Posamentier, 2002). The lower boundary of seismic facies L is developed on an unconformable and extremely irregular surface, on top the seismic facies R. Based on boreholes stratigraphy, the seismic facies L consists of fine-grained deposits.

**Seismic facies Ls.** Seismic facies Ls is evidenced by inclined oblique reflectors, characterized by discontinuous to chaotic low-amplitude reflections, which can terminate against each other. This reflectors configuration could resemble the typical pattern of Lateral Accretion Package (LAP) (Posamentier, 2002). High-amplitude reflections mark the bounding surfaces of these features giving to this facies a "U-shaped" geometry. This facies is scattered and is always contained within the seismic facies L. Concave upward reflectors are confined in the central portion of the basin, which probably are referable to channelized bodies whose location slightly shifted laterally (Deptuck et al., 2008; Schwab et al., 2007). Channel-fill facies mainly consist of coarse-grained deposits (generally sands) with minimal internal grading, as evidenced by boreholes stratigraphy (Fig. 4).

**Seismic facies BC.** Seismic facies BC is typically characterized by a chaotic pattern, with continuous, low-amplitude irregular reflectors. This facies is distributed in the upper part of the Corso section. Its basal boundary is highly irregular, and it was recognized down to ~80 ms. Both the stratigraphic position and the internal geometry of this seismic facies, coupled with the borehole stratigraphy, allow the correlation of this seismic facies with the chaotic breccias of CMA (Cosentino et al., 2017).



**Tomography**

For the investigation of the upper part of the Corso section, we interpreted the refraction tomography of the seismic section taking in to account several 30-m deep boreholes. The calculated Vp for the tomography are different from that used for the reflection profile (Tab. 2), depending on the different technique to elaborate seismic data (Improta et al., 2012). Then, for the

geological interpretation of the tomography, we used the trend of the Vp contour lines and the relative change of Vp. The interpretation allows to better define the unconformable boundary between the anthropic cover (AC) and the underlying Quaternary units, as well as the boundaries among COM, CMA, and MDS. Moreover, it clears the occurrence of the tectonic boundary due to the PDF and the presence of alluvial deposits within the calcareous breccias (CMA). The last has been confirmed by the stratigraphy of 3 boreholes, (Fig. 6).

**5 Discussion**

**Geological interpretation of the Corso section seismic reflection profile**

The seismic facies recognition along the Corso section, combined with previous studies on ASB geology (Mancini et al., 2012; Tallini et al., 2012; Cosentino et al., 2017; Nocentini et al., 2017) and the stratigraphy of the several boreholes drilled in the area after the April 6, 2009 Mw 6.1 L'Aquila earthquake, allowed to reconstruct a detailed geological model for AD (Fig. 7).

The semi-ubiquitarian presence in the examined boreholes (all except S12 and S13) of breccia deposits pertaining to the CMA (Fig. 3), which show different thickness, combined to the highly chaotic seismic facies BC permitted the recognition of this upper Middle Pleistocene unit for the upper part of the Corso section. In the high-resolution seismic profile, the younger covers (i.e. anthropic, alluvial, and colluvial deposits of Upper Pleistocene and Holocene), which are characterized by a few meters of thickness (Fig. 3), are indistinguishable from the underlying CMA.

The seismic facies L is mainly represented by continuous and parallel reflectors specific of fine-grained deposits. Based on this assumption and borehole stratigraphy, this seismic facies L corresponds mainly to MDS (Lower Pleistocene, Calabrian). The overlaying FGS (lower Middle Pleistocene), composed largely of silty sands, is possibly comprised in this seismic facies L, because it is not lithologically distinguishable from MDS. Moreover, the FGS mainly outcrops in the SW sector of AD, thus it is not ubiquitarian in the Corso section.

Seismic Facies Ls probably corresponds to channelized bodies of coarse-grained deposits within the generally finer MDS. Their presence is testified by sandy levels in the middle part of S5 well log (Fig. 4). These channels, sedimentologically defined as channel-point bars, are formed by dunes and ripple migration lateral to the currents, and are encased in floodplain pelite. The cross-bedded sands constitute mainly unidirectional flow bedload of a fluvial-channel fill. Similar channelized facies is also recorded in the LAQUI-CORE borehole, showing 31.4 m of gravels with sandy matrix (Porreca et al., 2016).



The seismic facies R is characterized by weakly continuous and divergent reflectors correlated to clastic coarse-grained deposits. In agreement with the interpretation of the Pettino I seismic reflection profile (Cosentino et al., 2017), this seismic facies could be linked to the CCF. Nevertheless, the CCF does not crop out close to AD (Nocentini et al., 2017).

Finally, the seismic facies S, although poorly definable, is correlated to the Meso-Cenozoic bedrock that, below AD, could be represented by Miocene carbonate-ramp as testified by the borehole S12 (Fig. 3).

**Tectonic features**

In a tectonically active intermontane continental basin, the evolution of infilling deposits depends on the subsidence of the basin, which is mainly controlled by the geometry of the fault systems affecting the basin (Gawthorpe and Leeder, 2000). ASB is considered a highly seismic region, in which the activity of the Plio-Quaternary extensional tectonics is recorded by several earthquake-induced features (Tallini et al., 2012; Storti et al., 2013).

In the Corso section, several normal faults can be detected (Fig. 7). They mainly cut the MDS and the underlying units (CCF and the Meso-Cenozoic bedrock). PDF is a SW-dipping normal fault characterized by a listric shape and different displacements. The resolution of the Corso section does not allow to evidence the displacement of the COM/CMA boundary (Fig. 7), although the COM thickness in the PDF hangingwall is higher than in the footwall, pointing to a possible tectonic activity of PDF during the Late Pleistocene (Fig. 6). Moreover, minor faults related to PDF do not displace the upper part of MDS testifying the syn-sedimentary faulting of MDS.

The syn-tectonic evolution of MDS is also suggested by the highly deformed laminations (convolute-like or deformed/broken laminae) recognizable at different depths in the borehole S5 (Fig. 4), which could represent soft-sediment deformations in response to palaeoearthquakes (paleoseismites) (Alfaro et al., 1997; Owen et al., 2011). This observation indicates strong seismic activity during the Calabrian within the ASB, as already suggested for the early Middle Pleistocene by Storti et al. (2013) on the base of soft-sediment deformations in the outcrops located nearby west of AD. Some empirical correlation between seismite type and paleoearthquake magnitude were proposed by Rodríguez-Pascua et al.(2000), but it requires the knowledge of the principal seismogenetic structures of the area. The simulated activation of the Pettino fault (PF) and its relative fault zones is expected to generate earthquakes with maximum magnitudes approaching Mw 6.7 (Galli et al., 2011; Moro et al., 2013). The soft-sediment deformations found in the borehole S5 are assimilable to "mushroom-shaped structure" due to silts protruding into laminites. This kind of structures are possibly caused by paleoearthquakes that range from Mw 5 to 7 (Rodríguez-Pascua et al., 2000), which are in accordance with paleomagnitudes revealed by Storti et al., (2013) from the same formation.

Using rough correlation between the faulted interval and the age of deposition of CMA (~300 ka) (Cosentino et al., 2017) a ~0.14mm/yr slip rate is inferred for PDF, which represents a similar value for other slip rate calculated for single splay of the nearby active Paganica Fault responsible for the April 6, 2009 L'Aquila earthquake (Galli et al., 2010).



### New subsurface data for a 3D model

In Fig. 8, the geological sections A-A' and B-B'-B'' represent the geological subsoil model of AD, which has been updated through the new geological data from the Corso section and the stratigraphy of the deep boreholes (Fig. 3). More precisely, the section A-A' has been drown by integrating the section in Nocentini et al. (2017) with data from the Corso section. The

Meso-Cenozoic bedrock is located maximum at 600 m b.g.l., and though it is the deepest value for the bedrock depth in ASB, it is in accordance with Meso-Cenozoic bedrock depth of other intermontane basins of central Italy as the Fucino Basin (Cavinato et al., 2002) and the Paganica-San Nicandro-Castelnuovo Basin (Civico et al., 2017).

The Colle Cantaro-Cave Formation (CCF), which is the earliest deposit of ASB, should be present at the base of the basin-filling succession (Centamore and Dramis, 2010). CCF shows different thicknesses along the geological sections (syn-rift

wedge) and, like in the Pettino I seismic reflection profile (Cosentino et al., 2017), it assumes an onlap configuration over the Meso-Cenozoic bedrock due to the tectonic activity of the normal faults that were bounding the ASB during the late Pliocene-early Pleistocene (late Piacenzian – Gelasian). The successive depositional event occurred in a more stable tectonic environment, with sub-horizontal deposits of MDS showing the deposition of similar thickness in the same time span throughout the geological sections. The FGS is scarcely represented along both the geological sections where it is only a few

meter-thick or completely missing. FGS crops out mainly on the southern slope of AD and is sometimes recognized in some boreholes. All the previous stratigraphic units are unconformably overlain by CMA (Nocentini et al., 2017). As stated before, the sedimentological characteristics of CMA point to huge events of detrital deposition through debris flow and rock avalanche with debris produced mainly by the erosion of the northern margin of ASB (Gran Sasso chain), possibly during a cold late Middle Pleistocene event (Cosentino et al., 2017). CMA is covered by the Collemaggio Synthem (COM) interpreted as

colluviated paleosols in the epikarst filling (Nocentini et al., 2017). COM is barely visible in the Corso section, but is well recognizable in the refraction seismic tomography (Fig. 6) with maximum 20 meters-thick. Finally, alluvial, colluvial and anthropic filling (ALL) are imaged only in the seismic tomography profile and represent the present-day deposition of sediments in the current geological setting.

The highest thickness of ASB Plio-Quaternary deposits is placed in correspondence with the S10 borehole, in the southern part

of AD that defines the ASB depocentre. The Plio-Quaternary depocentre is located within a minor NNW-SSE oriented graben, which in turn is contained into the main regional graben of the Pettino Fault system (PF) (Fig. 9).

Fig. 9 shows the main faults affecting the AD subsoil and the contour line of the gravimetric anomalies [modified from Nocentini et al. (2017) and Blumetti et al. (2002), respectively]. The reported faults are mainly extensional and/or transtensive, showing the main activity during the Quaternary. The faults that define the geometry of the Plio-Quaternary basin are oriented

NW-SE, E-W, and NNE-SSW. The faults trend follows the gravimetric contour lines. Positive gravimetric anomalies (red line) correspond to the Meso-Cenozoic carbonate and Upper Miocene terrigenous reliefs, while the negative ones (blue line) represent the Plio-Quaternary basin-fill. The NW-SE and E-W striking extensional faults bound the main graben (PF, CF, and VMF), which corresponds to the present Aterno River Valley, where the Plio-Quaternary continental deposits were

accumulated for the increase of the accommodation space due to the activity of the fault system associated with the master SW-dipping Pettino Fault. The principal local negative gravimetric anomaly (<-3.2 mGal) is exactly in AD and it is bounded by NW-SE normal faults (PF, CF, and VMF) and two NNE-SSW striking transtensive faults (SGF and SEF), which developed as transfer faults. These faults generate, within the main graben, a deep lowered sub-rectangular area placed exactly in AD,

characterized by a minor NNW-SSE graben bounded by TF and PDF faults. This subsurface tectonic setting is well represented by the two orthogonal geological section of Fig. 8. The extensional fault system of Fig. 9 follows a fractal-type and scale-independent arrangement with the normal faults, as a key structural elements, repeating at different scale (Ackermann et al., 2001; Turcotte, 1997).

The contour lines of the top of the Meso-Cenozoic bedrock permit to highlight the tectonic morphology of the lowered sub-

10 rectangular area (Fig. 10). The contour lines show mainly a NW-SE oriented enclosed area, with the maximum depth collocated right in the southern part of AD. In terms of local seismic response, the downthrown sub-rectangular block of bedrock highlights a deep basin geometry in NNW-SSE and E-W directions, evidencing a bi-dimensional or possibly three-dimensional seismic effects that must be considered for the mitigation of the seismic hazard of the significant cultural heritage urban area of L'Aquila downtown.

## 6 Conclusion

The geological interpretation of the ca. 1000 m-long high-resolution seismic reflection profile (Corso section), supported by deep boreholes and refraction tomography, allowed to greatly update the geological subsoil model of L'Aquila downtown (AD) and to investigate the Plio-Quaternary evolution of L'Aquila-Scoppito Basin (ASB).

The main results are as follows:

1.   the available seismic and borehole data permitted to reconstruct the AD subsoil model in two orthogonal directions highlighting a quasi tri-dimensional model (Figs. 8 and 10).

   2.   the major thickness of the Plio-Quaternary succession of the basin fill below AD was recognized with a maximum thickness of 600 m. It corresponds to the main depocentre of ASB located in the southern part of the Corso section (Figs. 5, 8, and 10).

3.   the ASB depocentre is tectonically located within a minor NNW-SSE oriented graben, in turn, contained into a main regional graben related to the tectonic activity of the Pettino Fault. This observation suggests that the active tectonics and seismicity of intermontane basins of central Italy, such as ASB, are related to Plio-Quaternary extensional fault systems characterized by complex graben structure from regional to local scales (Fig. 9).

   4.   the S10 borehole and the Corso section testify a thickness of at least 400 m for Madonna della Strada System (MDS).

As a classical flood plain deposit, MDS presents within fine-grained layers also some channelized bodies observable both in the Corso section and in the S5 borehole stratigraphy (Fig. 5).

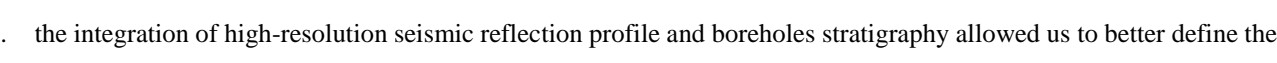

5. the integration of high-resolution seismic reflection profile and boreholes stratigraphy allowed us to better define the Plio-Quaternary tectono-stratigraphic evolution of ASB and the possible post-CMA (i.e., post late Middle Pleistocene) activity of the Piazza Duomo Fault.

The results were useful to reconstruct a detailed geological subsoil model at urban scale, and thus to provide indication
on the seismic site effects and on the activity of buried faults. This information was essential to design urban masterplan able to mitigate the Seismic Hazard of cultural heritage cities of central Italy, such as the case study site of L'Aquila downtown.

**Acknowledgements.**

We would like to thank the CARISPAQ institution to grant funds for the seismic reflection survey investigation. Thanks are
also extended to all the IAMC-CNR people who helped us during the geophysical acquisition phase in the field. The work strongly benefited from preliminary discussion on the seismic profile interpretation with Massimiliano Rinaldo Barchi and Massimiliano Porreca (Perugia University) who are warmly thanked. This work was carried out within the general agreement between L'Aquila University (Dipartimento di Ingegneria Civile, Edile-Architettura e Ambientale) and Roma Tre University (Dipartimento di Scienze). The grant to Dipartimento di Scienze, Università degli Studi di Roma Tre (MIUR-Italy Dipartimenti
di Eccellenza, articolo 1, commi 314 – 337 legge 232/2016) is gratefully acknowledged.

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



**Figure 1: Simplified structural sketch of the central Apennines. A: map of the Seismic Hazard in Italy, showing peak ground accelerations (g) that have a 10% chance of being exceeded in 50 yr (Meletti and Montaldo, 2007). B: main Pliocene–Quaternary intermontane basins of the central Apennines. 1-Quaternary volcanic rocks; 2-Plio–Quaternary marine and continental deposits; 3-Neogene foredeep turbiditic deposits; 4- Meso-Cenozoic carbonate (platform, slope and basin) deposits; 5-major thrust; 6-normal fault; PF: Pettino Fault; SPF: Scoppito-Preturo Fault; VMF: Via Mausonia Fault; PAF: Paganica Fault; ASB: L'Aquila-Scoppito Basin. Modified from Cosentino et al., 2017.**





**Figure 2: Geologic map of the studied area (modified from Nocentini et al., 2017). PF: Pettino Fault; CF: Castello Fault, SGF: San Giuliano Fault; TF: Tribunale Fault; PDF: Piazza Duomo Fault; SEF: Sant'Elia Fault; VMF: Via Mausonia Fault.**

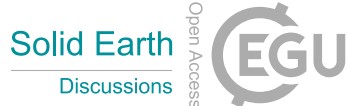

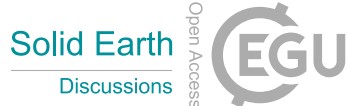

**Figure 3: Correlation panel of selected borehole stratigraphy with location of boreholes. For the legend see Fig. 2.**



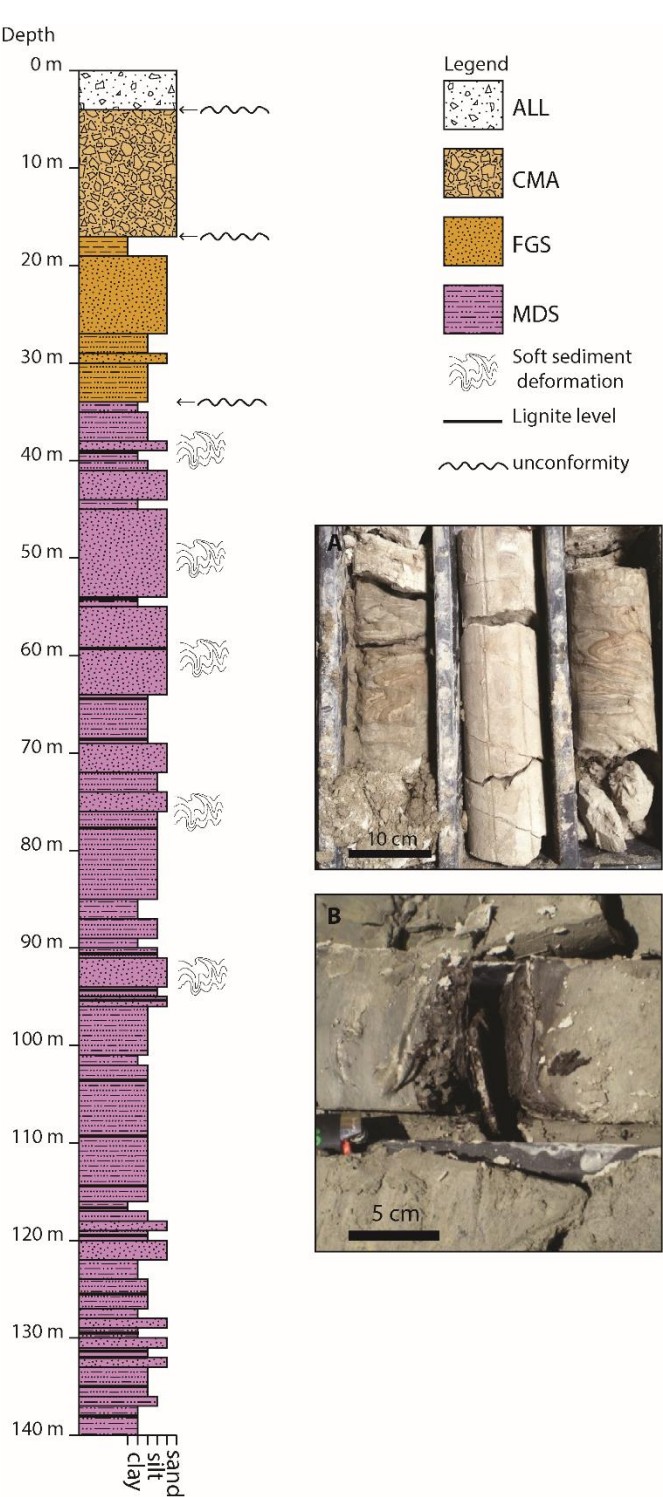

**Figure 4: Detailed borehole stratigraphy of S5. A: detail of soft sediments deformation (possibly seismite) recognized at ~77 m of depth; B: lignite level in the lower part of S5 borehole.**





**Figure 5: A) 2D depth-migrated reflection of the Corso section (horizontal scale= common deep point, vertical scale= two-way time); B) line drawing of seismic facies (following tab. 2), horizons, and faults (horizontal scale= linear meters, vertical scale= two-way time); C) line drawing of the seismic profile with recognised seismo-stratigraphic units (horizontal scale= linear meters, vertical scale= two-way time). Legend: 1= seismic facies BC; 2= channelized bodies in seismic facies BC; 3= seismic facies L; 4= seismic facies Ls; 5= seismic facies R; 6= seismic facies S; 7= fault; 8= channelized bodies; 9= unconformity; 10= top of Meso-Cenozoic bedrock; PDF: Piazza Duomo Fault.**





**Figure 6: Seismic tomography of the Corso section interpreted through borehole stratigraphies. A: uninterpreted seismic refraction tomography. B: seismic refraction tomography with the seismic rays. C: interpreted seismic refraction tomography with the shallow borehole stratigraphies. D: geological interpretation of the seismic refraction tomography AC: anthropic cover (buried foundation and anthropic fill characterized by high and low Vp, respectively); COM: reddish colluviated fine-grained paleosols (Upper Pleistocene); CMA: calcareous breccia and gravel (upper Middle Pleistocene); CMA-f: alluvial sand, pelite and gravel (upper Middle Pleistocene); MDS: alluvial pelite and sand (Calabrian p.p.). In the upper part of this unit, the possible presence of thin alluvial deposits referring to FGS (lower Middle Pleistocene) cannot be ruled out; PDF: Piazza Duomo Fault.**



**Figure 7: Geological interpretation of the Corso section seismic profile. For the legend see Fig. 2**





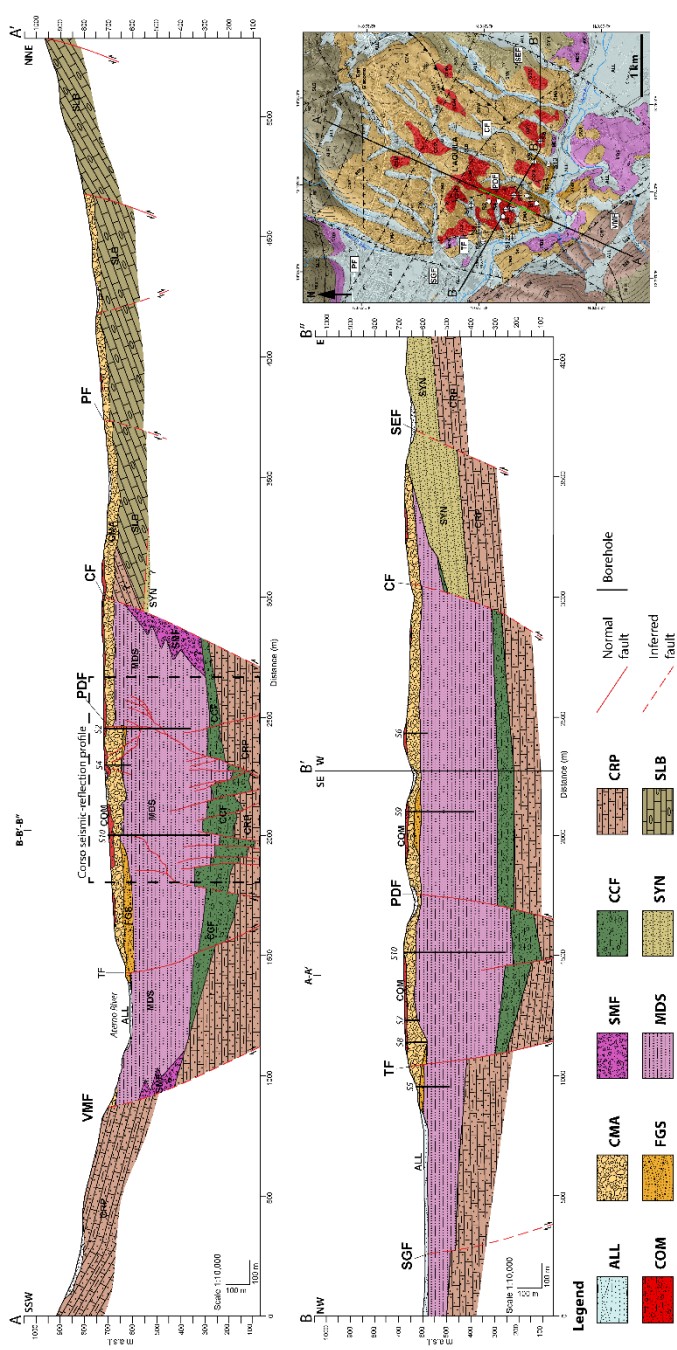

**Figure 8: Geological sections crossing L'Aquila hill (modified from Nocentini et al., 2017). The acronyms of the faults are: PF: Pettino Fault; CF: Castello Fault, SGF: San Giuliano Fault; TF: Tribunale Fault; PDF: Piazza Duomo Fault; SEF: Sant'Elia Fault; VMF: Via Mausonia Fault. For both the location and legend see Fig. 2.**





**Figure 9: Tectonic and gravimetric sketch of Quaternary L'Aquila-Scoppito Basin (modified from Nocentini et al., 2017 and Blumetti et al., 2002). 1: main normal and transtensive fault (PF: Pettino Fault, CF: Castello Fault, VMF: Via Mausonia Fault, SGF: San Giuliano Fault, SEF: Sant'Elia Fault); 2: minor normal fault (PDF: Piazza Duomo Fault, TF: Tribunale Fault); 3: minor right lateral faults; 4: Quaternary deposit; 5: Meso-Cenozoic bedrock; 6: contour lines of residual gravimetric anomalies in mGal (+ and - : positive and negative anomaly, respectively); 7: trace of the geological sections reported in Fig. 8; 8: L'Aquila downtown (AD).**





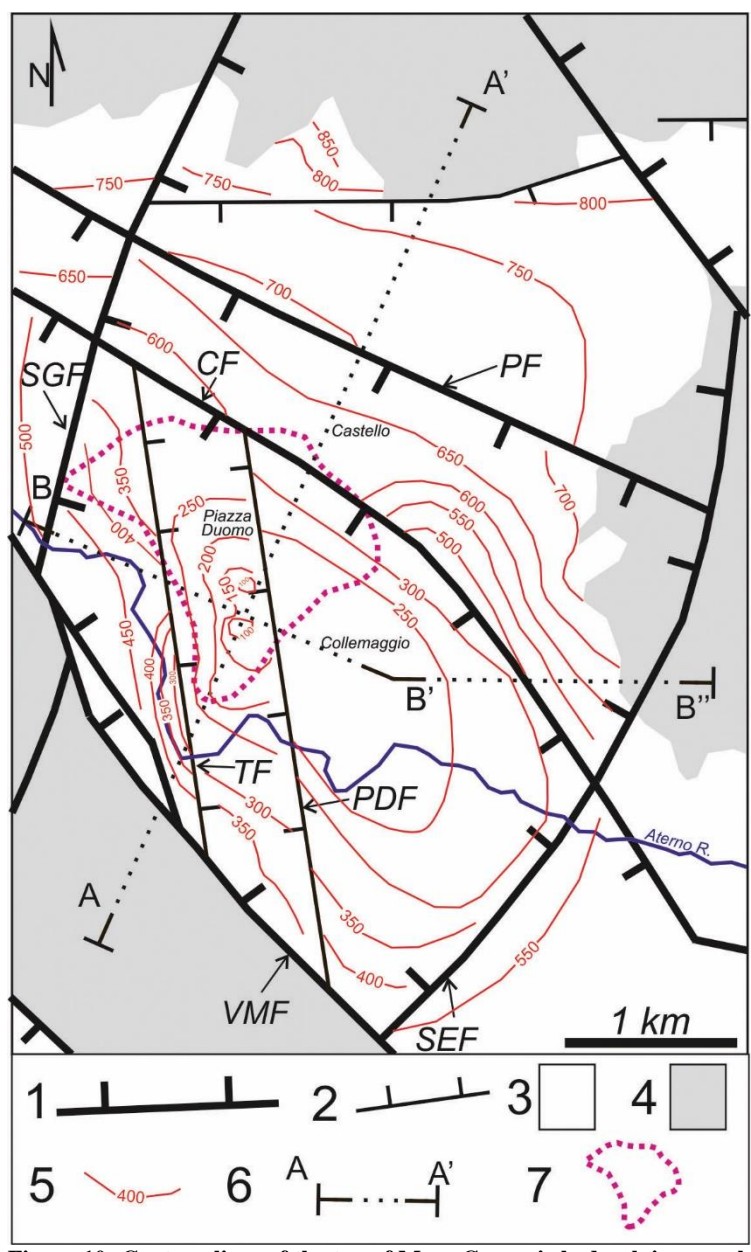

**Figure 10: Contour lines of the top of Meso-Cenozoic bedrock in m a.s.l. 1: main normal and transtensive fault (PF: Pettino Fault, CF: Castello Fault, VMF: Via Mausonia Fault, SGF: San Giuliano Fault, SEF: Sant'Elia Fault); 2: minor normal fault (PDF: Piazza Duomo Fault, TF: Tribunale Fault); 3: Quaternary deposit; 4: Meso-Cenozoic bedrock; 5: contour lines of the top of Meso-Cenozoic bedrock in m a.s.l.; 6: trace of the geological sections reported in Fig. 8; 7: L'Aquila downtown (AD).**



**Table 1: Acquisition characteristics of the Corso section.**

|  | **High-resolution of the Corso section** |
|---|---|
| Length | 960 m |
| Seismic Source | IVI - MINIVIB |
| Geophone interval | 5 m |
| Shot spacing | 10 m |
| Number of shots | 91 |
| Receiver Spread size | 192 |
| **Recording parameters** |  |
| Sweep characteristics | Linear, 3 x15 sec up-sweep from 10 to 200 Hz |
| Sampling rate | 1 ms |
| Record Length | 15 s |
| Geophone | 10 Hz |



**Table 2: Vp-lithology associations and their possible equivalent formations used for the interpretation of the Corso section.**

| Seismic facies | Representation | Vp range (m/s) | Lithology | Geological interpretation |
|---|---|---|---|---|
| BC | | 2250-2500 | Sands and conglomerates, breccias | Fan deposits and slope breccias |
| Ls | | 1500-2000 | Sands | Channelized deposits |
| L | | 1500-2000 | Clay, silts and sands | Alluvial plain deposits |
| R | | 2250-3000 | Sands and conglomerates, breccias | Fan deposits and slope breccias |
| S | | > 3250-3500 | Slope to basin carbonate sequences (Meso-Cenozoic bedrock) | Meso-Cenozoic bedrock |