# Peer review of "Seismic Hazard of L'Aquila downtown (central Italy): new insights for 3D geological model based on high-resolution seismic reflection profile and borehole stratigraphy"

_Solid Earth, 2019_

## Referee Comment (RC1) · Anonymous Referee #1 · 16 Mar 2019

Dear Editor, I have completed reviewing this manuscript evaluating seismic hazard L'Aquila in Italy based on integrated geological and geophysical data which might be a state of the art approach for other historical downtowns in Italy and also other extensional regions such as Greece and western Turkey. One of the impressed points of this manuscript is showing the significance of field-based local and regional geological information to better understand seismic hazard for a very specific and fine-scale problem which is crucial for the society. The language of the paper is very good and easy to follow and it is well structured. The only thing that I would like to request from

authors providing a photo from L'Aquila downtown to readers. I visited the town but other readers to whom may not be visited it might be very useful to keep this paper in their mind. Yours Sincerely
* * *

---

## Referee Comment (RC2) · Anonymous Referee #2 · 19 Mar 2019

**General comments**

I have read your paper with great interest and I think it is a very nice case study about the seismic hazard of downtown L'Aquila based on reflection seismic and borehole stratigraphy. You present an improved geological model for the investigation area and you have connected your results to the evolution of the L'Aquila-Scoppito Basin and the seismic hazard of this specific region. Overall the article is well structured and the topic (structural geology and geophysics, in the context of seismic hazard) is relevant for Solid Earth. I therefore recommend it for publication with revision. Although the English

grammar could be improved and you should avoid long sentences which consist of four or even more lines. Shorter sentences will make it easier to read and understand the manuscript.

I have a few comments which are suggestions that I hope may help in improving the quality of the paper.

**Specific comments**

page 2 line 6: "Amplification effect related to the seismic wave propagation..." - This is a very important aspect regarding this study and therefore should be explained in more detail and you should include more recent literature.

page 2 line 15: In this part you are describing how important it is to use different geological and geophysical methods in order to get a reliable 3D model of the underground, but the cited paper deals only with shear-wave velocity profiles and ambient vibration array measurements. You should cite more papers in the context of 3D modelling that deal with the other geological and geophysical methods that you mention.

page 3 line 4: I suggest to write the abbreviations for the Scoppito-Preturo normal fault (SPF) and the Pettino normal fault (PF) in brackets as you have done for the geological formations. This makes it easier for the reader to find them in the corresponding figures.

page 3 line 5: In the text you are referring to figure 1, but in fact you are only describing figure 1b. Figure 1a shows the peak ground acceleration, which is not explained in the text. Please correct this.

page 4 line 8: Instead of "capped" I suggest to use the word "covered".

page 4 line 29: I guess that you stacked the entire recorded signal to improve the

signal-to-noise ratio, and not just the sweeps. Please correct.

page 5 line 2: In seismic literature it is common to write "a maximum CMP fold of 48 traces" and not "4800%", because the fold of the stack is determined by the number of traces in the CMP gather.

page 5 line 10: "Tomography data was used both to extend the seismic imaging..". A seismic tomography shows velocity anomalies which do not necessarily correspond to structural features. Therefore, a reader, who is not familiar with seismic techniques, might misunderstand this part. I suggest to write one or two explanatory sentences.

page 5 line 10 to 11: "...very near surface (first 30-50 m) since usually this part is not sampled, even by "shallow" seismic reflection techniques...". The statement that the first 30 to 50 m cannot be imaged even by shallow reflection seismic techniques is incorrect. Many studies, especially from the last 10 years, have shown the successful application of shear wave (SH-wave) reflection seismic to image the very-near surface in high-resolution, sometimes less than 1 m. I strongly suggest you should read some publications dealing with shear-wave reflection seismic for near-surface applications and than change this part in your text. In the following, I listed several publications which might be helpful to you:

Beilecke, T., Krawczyk, C.M., Tanner, D.C. & Ziesch, J.: Near-surface fault detection using high- shear wave reflection seismics at the CO2CRC Otway Project site, Australia, Journal of Geophysical Research: Solid Earth, 121, 1–23, doi = 10.1002/2015JB012668, 2016.

Harris, J.B.: Application of shallow shear-wave seismic reflection methods in earthquake hazard studies, The Leading Edge, 29, 8, 960-963, doi = 10.1190/1.3480010.

Kammann, J., Hübscher, C., Boldreel, L.O. & Nielsen, L.: High-resolution shear-wave seismics across the Carlsberg Fault zone south of Copenhagen — Implications for

linking Mesozoic and late Pleistocene structures, Tectonophysics, 682, 56-64, doi = 10.1016/j.tecto.2016.05.043.

Krawczyk, C.M., Polom, U., Trabs, S. & Dahm, T.: Sinkholes in the city of Hamburg-New urban shear-wave reflection seismic system enables high-resolution imaging of subrosion structures, J. Appl. Geophys., 78, 133–143, doi = 10.1016/j.jappgeo.2011.02.003, 2012.

Krawczyk, C.M., Polom, U. & Beilecke, T.: Shear-wave reflection seismics as a valuable tool for near-surface urban applications, The Leading Edge, 32, 3, 256–263, doi = 10.1190/tle32030256.1, 2013.

Polom, U., Bagge, M., Wadas, S., Winsemann, J., Brandes, C., Binot, F. & Krawczyk, C.M.: Surveying near-surface depocentres by means of shear wave seismics, First Break, 31, 8, 67–79, 2013.

Pugin, A.J.-M., Brewer, K., Cartwrigh, T., Pullan, S.E., Didier, P., Crow, H. & Hunter, J.A.: Near surface S-wave seismic reflection profiling—new approaches and insights, First Break, 31, 49–60, 2013.

Pugin, A.J.-M., Pullan, S.E. & Hunter, J.A.: Shear-wave high-resolution seismic reflection in Ottawa and Quebec City, Canada, The Leading Edge, 32, 3, 250–255, doi = 10.1190/tle32030250.1, 2013.

Wadas, S.H., Tanner, D.C., Polom, U. & Krawczyk, C.M.: Structural analysis of S-wave seismics around an urban sinkhole; evidence of enhanced dissolution in a strike-slip fault zone, Natural Hazards and Earth System Sciences, 17, 2335–2350, doi = 10.5194/nhess-17-2335-2017, 2017.

From my personal view and based on your seismic results, I think it would be very interesting to carry out an SH-wave reflection seismic profile in downtown L'Aquila, because it could deliver very promising results regarding the internal structures of the sedimentary infill and the detection of hidden near-surface faults. Maybe this would be

a nice topic for a future project.

page 5 line 18: "...less than the average travel time pick error." - What exactly is the average travel time pick error? Give a number.

General comments and or questions to the paragraph 'Seismic data acquisition':

What kind of filters were used during data processing?

Can you say anything about signal attenuation? The sweep frequency was 10 to 200 Hz, but what was the frequency of the recorded signal.

To get a better insight into the quality of the seismic data, show some records. It would be very helpful, e.g. before/after comparison for some processing steps like amplitude correction, filtering or deconvolution.

page 6 line 16 to 17: "In the S5 borehole, a pedogenetic horizon (oxidized surface) distinguishes the FGS from the fine-grained deposits referable to MDS." - In the corresponding figure 4 you have described the boundary between FGS and MDS as an "unconformity". Please use the same terms in the text and the figures. Using different terms might confuse the reader.

page 7 line 5: use "deepest part" instead of "deepest portion".

page 7 line 30: "Its basal boundary is highly irregular, and it was recognized down to 80 ms." - In the section 'Seismic data acquisition' you have written that the first 50 m were not properly imaged by the reflection seismic due to the resolution limits in the near-surface. Are you sure that you can get a reliable interpretation for facies BC that is located in the uppermost part of your seismic profile?

page 8 line 3: "The calculated Vp for the tomography are different from that used for the reflection profile.." - What exactly is the difference between both velocity fields?

Give numbers or show an image in which both velocity fields are compared.

page 8 line 15 and line 24: "ubiquitarian" - I guess you mean ubiquitous. The meaning of ubiquitarian is: relating to or believing in the doctrine that Christ is present every-where at all times. I guess that is not what you wanted to say.

page 9 line 8 to 9: "...the evolution of infilling deposits depends on the subsidence of the basin, which is mainly controlled by the geometry of the fault systems affecting the basin." - What is with subsidence resulting from the accumulation of large volumes of sediments? In many basins we have an interaction between fault-related and loading-related subsidence.

page 10 line 4: "...has been drown..." - I guess you mean "has been drawn".

page 10 line 17: "...the sedimentological characteristics of CMA point to huge events of..." - a short repetition of the most important sedimentological characteristics of CMA would be helpful at this point.

page 10 line 21 to 23: see my comment on seismic tomography and structural inter-pretation for page 5 line 10.

General comments and or questions to the paragraph 'Discussion':

The discussion of the tectonic features and the subsurface model is good, but what is completely missing so far is the critical discussion of the reflection and refrac-tion seismic methods. Most of your results, except for the borehole stratigraphy, are based on geophysics. As a consequence you should discuss problems and disad-vantages/advantages of both methods regarding data acquisition and processing. For example, I think discussing the resolution limits of your data would be very helpful. You should ask yourself 'What could have been done better and what other geophysical investigations would I carry out in the future, in the case of a subsequent project.

Other questions you could think of are 'Can you compare your results with other in-termontane basins in seismically-active regions?' and 'With your results is it possible to better estimate the future seismic hazard for downtown L'Aquila? For example, can you define particular risk areas, where damage to buildings and infrastructure would be higher than in other areas due to the local near-surface geology derived from your data.

**Technical corrections**

Figure 4: The marked "lignite level" (black line) in the stratigraphy plot is hardly visible. Maybe using a different colour, e.g. red, would be better.

Figure 5: For a better correlation of seismic and borehole results it would be nice if you could draw the location of the nearby boreholes into the seismic profile. This will help to better verify the seismic interpretation.

Figure 5: Why have you abbreviated the seismic facies twice? In the text and in table 2 where you are describing the seismic facies in detail you use BC, Ls, L, R and S as abbreviations for your seismic facies analysis. In the figure and the figure caption you use the following legend "1= seismic facies BC; 2= channelized bodies in seismic facies BC; 3= seismic facies L; 4=seismic facies Ls; 5= seismic facies R; 6= seismic facies S". I understand why you had to find new abbreviations for e.g. the channel bodies in seismic facies BC, but I do not understand why you had to rename the facies classes themselves. This would be very confusing for the reader. I would rewrite the legend in the figure caption and the corresponding part of figure 5c like this: BC = fan deposits and slope breccias; 1 = channelized bodies in seismic facies BC; L = alluvial plain deposits; 4 = channelized deposits; R = fan deposits and slope breccias; S = meso-cenozoic bedrock; 2= fault; 3= channelized bodies; 4= unconformity; 5= top of Meso-Cenozoic bedrock. This way you do not have two different abbreviations for the same facies.

Figure 5: In the text you have written that you used a Kirchhoff time migration, but in the figure caption you have written that figure 5a shows the "2D depth-migrated reflection of the Corso section". Have you carried out time-migration or depth-migration? You have also written "common deep point" but it must be 'common depth point' and instead of "two-way time" you should use 'two-way traveltime'.

Figure 7: When drawing faults into a cross section it is necessary to draw arrows indicating the fault movement.

Overall, a very nice work, congratulations!

---

## Author Comment (AC1) · 12 Apr 2019

Dear Editor, I have completed reviewing this manuscript evaluating seismic hazard L'Aquila in Italy based on integrated geological and geophysical data which might be a state-of-the-art approach for other historical downtowns in Italy and also other extensional regions such as Greece and western Turkey. One of the impressed points of this manuscript is showing the significance of field-based local and regional geological information to better understand seismic hazard for a very specific and fine-scale problem which is crucial for the society. The language of the paper is very good and

easy to follow and it is well structured. The only thing that I would like to request from authors providing a photo from L'Aquila downtown to readers. I visited the town but other readers to whom may not be visited it might be very useful to keep this paper in their mind. Yours Sincerely

Answer: many thanks for your suggestions and your kind comment on our manuscript. Following your suggestion, we add and upload a new Fig. 2 concerning the Google map and landscape of L'Aquila city.
* * *
[Figure]

**Fig. 1.** Figure 2: a) L'Aquila view from Google Earth; the red line corresponds to the high-resolution reflection seismic profile (Corso Section); the yellow dotted line corresponds to medieval wall. b) L'Aqui

---

## Author Comment (AC2) · 12 Apr 2019

General comments I have read your paper with great interest and I think it is a very nice case study about the seismic hazard of downtown L'Aquila based on reflection seismic and borehole stratigraphy. You present an improved geological model for the investigation area and you have connected your results to the evolution of the L'Aquila-Scoppito Basin and the seismic hazard of this specific region. Overall the article is well structured and the topic (structural geology and geophysics, in the context of seismic hazard) is relevant for Solid Earth. I therefore recommend it for publication with revision. Although

the English grammar could be improved and you should avoid long sentences which consist of four or even more lines. Shorter sentences will make it easier to read and understand the manuscript. I have a few comments which are suggestions that I hope may help in improving the quality of the paper.

Answer: we check the language along the text, simplifying the long sentences.

Specific comments page 2 line 6: "Amplification effect related to the seismic wave propagation..." - This is a very important aspect regarding this study and therefore should be explained in more detail and you should include more recent literature.

Answer: we add the following new text and new references taking in account your suggestion: "Amplification effects related to the seismic wave propagation are mainly due to vertical and lateral changes in thickness and mechanical behaviour of the subsoil lithological units and/or abrupt variation in topography (Bard and Gariel 1986; Lee et al. 2009; Marzorati et al. 2011). The amplification effects are analysed with a monodimensional or bidimensional numerical approach in parallel layering or in sedimentary basin and topographic relief respectively. In the last decades, the 2D seismic amplification due to the sedimentary basins such as the alluvial and intermontane basins, were specifically studied because of the large presence of cities and infrastructures. The 2D seismic effects can be related to the confinement into the basin of S and surface waves produced on the bedrock-soil boundary, because of constructive interference between reflected and refracted waves (Semblat et al. 2005; Pilz et al. 2018)." New references: Lee, S., Komatitsch, D., Huang, B. and Tromp, J. Effects of Topography on Seismic-Wave Propagation: An Example from Northern Taiwan, Bull. Seism. Soc. of America, 99(1), 314–325, 2009. Marzorati, S., Ladina, C., Falcucci, E., Gori, S., Saroli, M., Ameri, G., and Galadini, F.: Site effects "on the rock": the case of Castelvecchio Subequo (L'Aquila, central Italy). Bull. Earth. Eng., 9(3), 841-868, 2011. Semblat, J. F., Kham, M., Parara, E., Bard, P. Y., Pitilakis, K., Makra, K., and Raptakis, D.: Seismic wave amplification: Basin geometry vs soil layering. Soil dynamics and earthquake engineering, 25(7-10), 529-538, 2005 Pilz, M., Parolai, S., Petrovic, B., Silacheva,

N., Abakanov, T., Orunbaev, S., and Moldobekov, B. Basin-edge generated Rayleigh waves in the Almaty basin and corresponding consequences for ground motion amplification. Geophysical Journal International, 213(1), 301-316, 2017.

page 2 line 15: In this part you are describing how important it is to use different geological and geophysical methods in order to get a reliable 3D model of the underground, but the cited paper deals only with shear-wave velocity profiles and ambient vibration array measurements. You should cite more papers in the context of 3D modelling that deal with the other geological and geophysical methods that you mention.

Answer: following your suggestion we add new references: Carrasco, R. M., Turu, V., Pedraza, J., Muñoz-Martin, A., Ros, X., Sanchez, J., Ruiz-Zapata, B., Olaiz, A. J., and Herrero-Simon, R. : Near surface geophysical analysis of the Navamuño depression (Sierra de Bejar, Iberaina Central Sysntem): Geometry, sedimentary infill and genetic implication of tectonic and glacial footprint, Geomoprhology, 315, 1-16, 2018. Civico, R., Sapia, V., Di Giulio, G., Villani, F., Pucci, S., Baccheschi, P., Amoroso, S., Cantore, L., Di Naccio, D., Hailemikael, S., Smedile, A., Vassallo, M., Marchetti, M. and Pantosti, D.: Geometry and evolution of a fault-controlled Quaternary basin by means of TDEM and single-station ambient vibration surveys: The example of the 2009 L'Aquila earthquake area, central Italy, J. Geophys. Res. Solid Earth, 122(3), 2236–2259, doi:10.1002/2016JB013451, 2017. Maresca, R., and Berrino, G. Investigation of the buried structure of the Volturara Irpina Basin (southern Italy) by microtremor and gravimetric data. Journal of applied geophysics, 128, 96-109, 2016

page 3 line 4: I suggest to write the abbreviations for the Scoppito-Preturo normal fault (SPF) and the Pettino normal fault (PF) in brackets as you have done for the geological formations. This makes it easier for the reader to find them in the corresponding figures.

Answer: done.

page 3 line 5: In the text you are referring to figure 1, but in fact you are only describing figure 1b. Figure 1a shows the peak ground acceleration, which is not explained in the

text. Please correct this. Answer: We modified the text as follows: For the evaluation of the seismic local effect and the recognition of the active faults, the knowledge of the 3D geological model is primary to mitigate the Seismic Hazard of cultural heritage cities of central Italy, which are mainly placed in Plio-Quaternary intermontane basins characterised by high seismicity, as demonstrated by the peak ground acceleration (Fig. 1A) (Meletti and Montaldo, 2007) and by the recent earthquakes (Fig. 1B) (i.e. Mw 6.1 L'Aquila event of April 6, 2009 and Mw 6.0 Amatrice event of August 24, 2016: Gruppo di Lavoro MS–AQ, 2010; Rossi et al., 2019).

page 4 line 8: Instead of "capped" I suggest to use the word "covered".

Answer: done.

page 4 line 29: I guess that you stacked the entire recorded signal to improve the signal-to-noise ratio, and not just the sweeps. Please correct.

Answer: we stacked the entire recorder signal and not just the sweeps. It was corrected as follows: "The geometry consists of a dense (5 m spacing) 192-channels 10-Hz vertical geophone array. The source move-up was 10 m; at each of the 91 vibration points, three 15 s long, 10-200 Hz sweeps were performed then we stacked the correlated data to improve the signal to noise ratio".

page 5 line 2: In seismic literature it is common to write "a maximum CMP fold of 48 traces" and not "4800%", because the fold of the stack is determined by the number of traces in the CMP gather.

Answer: done.

page 5 line 10: "Tomography data was used both to extend the seismic imaging.". A seismic tomography shows velocity anomalies which do not necessarily correspond to structural features. Therefore, a reader, who is not familiar with seismic techniques, might misunderstand this part. I suggest to write one or two explanatory sentences.

Answer: we have clarified in the text the issue of the different resolution between the

reflection seismic and the refraction seismic. In particular, in seismic tomography, we obtain a 2D Vp field recognizing very well lateral Vp variations. The seismic reflection allows to identify stratigraphic and structural discontinuities. Due to strong lateral Vp variation, the reflection imaging is difficult in shallow subsoil because only a small number of short-offset traces for any shot gather reflection signal. Moreover, the reflections are usually masked by strong noise which often is difficult to remove and therefore it produces a low quality of imaging. In the following the modifying text: "In particular, Fig. 5 shows the velocity analysis process performed by a CDP supergather. The CDP supergather creates and inserts into the flow, the trace sets composed of several CDP gathers (in 2D case) breaking the created set into trace groups with a specified constant offset step (binning), subsuming the binned traces within the set. Semblance function was used to estimate the Vp (RMS) coherence vs. TWT. Stacking velocity (gray line) was obtained by the picking on the maxima coherence points. Interval velocity (black line) was also obtained by Dix formula. Coherent points were localized at 95 ms (VRMS =2030 m/s), 235 ms (VRMS =1750 m/s), 440 ms (VRMS =2100 m/s), and 550 ms (Vp=3000 m/s). Fig. 6 shows a comparison before and after some processing steps. In particular, Fig. 6a shows two raw shot gathers and the corresponding gathers (Fig. 6b) after the application of some processing steps like amplitude correction, filtering and predictive deconvolution. Shot gathers of Fig. 6b show the strong attenuation of the multiples and of the ground roll. Fig. 7 instead shows the spectral sweep content (red line) and the spectral content of all the acquired traces (black line). After the 60 Hz occurred strong signal attenuation. In order to vertical and horizontal resolution in seismic reflection the vertical resolution can be calculated from the length of the propagation wave and the layer thickness below 1/4 wavelength for resolving limits of beds (Chopra et al., 2006); The Fresnel zone, indeed, defines horizontal resolution by the seismic signal at the certain depth. In particular, the First Fresnel Zone (FFZ) radius can be calculated by the formula (Chopra et al., 2006): $R = V/2 \sqrt{(t\_0/f)}$; where R is the FFZ ray. Table 2 reports the values of horizontal and vertical resolution calculated for the reflection data at 0.050 and 0.100 TWT. The Vp and F are referred to

the specific TWT as shown in the table." We also added new Figures 5-6-7-8 and new Table 2.

page 5 line 10 to 11: "...very near surface (first 30-50 m) since usually this part is not sampled, even by "shallow" seismic reflection techniques...". The statement that the first 30 to 50 m cannot be imaged even by shallow reflection seismic techniques is incorrect. Many studies, especially from the last 10 years, have shown the successful application of shear wave (SH-wave) reflection seismic to image the very-near surface in high-resolution, sometimes less than 1 m. I strongly suggest you should read some publications dealing with shear-wave reflection seismic for near-surface applications and than change this part in your text. In the following, I listed several publications which might be helpful to you: Beilecke, T., Krawczyk, C.M., Tanner, D.C. & Ziesch, J.: Near-surface fault detection using high- shear wave reflection seismics at the CO2CRC Otway Project site, Australia, Journal of Geophysical Research: Solid Earth, 121, 1–23, doi = 10.1002/2015JB012668, 2016. Harris, J.B.: Application of shallow shear-wave seismic reflection methods in earthquake hazard studies, The Leading Edge, 29, 8, 960-963, doi = 10.1190/1.3480010. Kammann, J., Hübscher, C., Boldreel, L.O. & Nielsen, L.: High-resolution shear-wave seismics across the Carlsberg Fault zone south of Copenhagen âAËŸT Implications for linking Mesozoic and late Pleistocene structures, Tectonophysics, 682, 56-64, doi =10.1016/j.tecto.2016.05.043. Krawczyk, C.M., Polom, U., Trabs, S. & Dahm, T.: Sinkholes in the city of Hamburg-New urban shear-wave reflection seismic system enables high-resolution imaging of subrosion structures, J. Appl. Geophys., 78, 133–143, doi = 10.1016/j.jappgeo.2011.02.003, 2012. Krawczyk, C.M., Polom, U. & Beilecke, T.: Shear-wave reflection seismics as a valuable tool for near-surface urban applications, The Leading Edge, 32, 3, 256–263, doi = 10.1190/tle32030256.1, 2013. Polom, U., Bagge, M., Wadas, S., Winsemann, J., Brandes, C., Binot, F. & Krawczyk, C.M.: Surveying near-surface depocentres by means of shear wave seismics, First Break, 31, 8, 67–79, 2013. Pugin, A.J.-M., Brewer, K., Cartwrigth, T., Pullan, S.E., Didier, P., Crow, H. & Hunter, J.A.: Near surface S-wave seismic reflection profilingâAËŸTnew approaches and insights, ËĞFirst

Break, 31, 49–60, 2013. Pugin, A.J.-M., Pullan, S.E. & Hunter, J.A.: Shear-wave high-resolution seismic reflection in Ottawa and Quebec City, Canada, The Leading Edge, 32, 3, 250–255, doi = 10.1190/tle32030250.1, 2013. Wadas, S.H., Tanner, D.C., Polom, U. & Krawczyk, C.M.: Structural analysis of Swave seismics around an urban sinkhole; evidence of enhanced dissolution in a strikeslip fault zone, Natural Hazards and Earth System Sciences, 17, 2335–2350, doi = 10.5194/nhess-17-2335-2017, 2017. From my personal view and based on your seismic results, I think it would be very interesting to carry out an SH-wave reflection seismic profile in downtown L'Aquila, because it could deliver very promising results regarding the internal structures of the sedimentary infill and the detection of hidden near-surface faults. Maybe this would be a nice topic for a future project.

Answer: thanks for the list of suggested publications. Since the seismic reflection is very difficult in urban areas and the shallow subsoil is characterized by strong lateral Vp variations, a strong scattering is generated and therefore it is difficult to have coherence of the reflected phases. We are convinced that the investigation in SH-wave could be very useful and therefore we consider it for a future project. We added some of the suggested publications in the introduction paragraph: Beilecke, T., Krawczyk, C.M., Tanner, D.C. & Ziesch, J.: Near-surface fault detection using high- shear wave reflection seismics at the CO2CRC Otway Project site, Australia, Journal of Geophysical Research: Solid Earth, 121, 1–23, doi: 10.1002/2015JB012668, 2016. Krawczyk, C.M., Polom, U., Trabs, S. & Dahm, T.: Sinkholes in the city of Hamburg-New urban shear-wave reflection seismic system enables high-resolution imaging of subrosion structures, J. Appl. Geophys., 78, 133–143, doi: 10.1016/j.jappgeo.2011.02.003, 2012. Krawczyk, C.M., Polom, U. & Beilecke, T.: Shear-wave reflection seismics as a valuable tool for near-surface urban applications, The Leading Edge, 32, 3, 256–263, doi: 10.1190/tle32030256.1, 2013.

page 5 line 18: "...less than the average travel time pick error." - What exactly is the average travel time pick error? Give a number. General comments and or questions

to the paragraph 'Seismic data acquisition': What kind of filters were used during data processing? Can you say anything about signal attenuation? The sweep frequency was 10 to 200 Hz, but what was the frequency of the recorded signal. To get a better insight into the quality of the seismic data, show some records. It would be very helpful, e.g. before/after comparison for some processing steps like amplitude correction, filtering or deconvolution.

Answer: the average travel time pick error was referred to RMS, which value is 5.65 ms. In order to your general comments and or questions to the paragraph 'Seismic data acquisition', we reported in the manuscript information concerning the processing on the pre-stack data using filtering, amplitude correction, predictive deconvolution, and signal attenuation. In the following the modifying text: "The model was adjusted until the misfit is minimized. The iterations were stopped when the RMS travel time residual (difference between the calculated travel times for the initial model and the observed ones) is 5.65 ms is less than the average travel time pick error. For refraction data analysis, all first-arrival travel times were accurately hand-picked on the common shot panels. Travel-time diagrams were created and checked for consistency, following the rules of Ackermann et al. (1986). The tomography resolution is connected to ray distribution into the cell and it has a different effect on the solution quality of the corresponding cell or model parameter. Fig. 8 shows the ray density distribution of the model. In order to evaluate the resolving power of the data set and to examine model resolution, we investigated various standard measures such as derivative weighted sum (DWS, Kissling, 1988). Table 3 reports the resolution parameters versus depth considering the cell size utilized in inversion process. Seismic tomography is used both to apply in a suitable way the static corrections of the reflection data and to have complementary information of the P wave velocity field. In fact, reflection imaging is difficult in shallow subsoil because only a small number of short-offset traces for any shot gather present reflection signal; and those reflections present are usually masked by strong coherent noise which must be strongly attenuated before imaging the reflections."

page 6 line 16 to 17: "In the S5 borehole, a pedogenetic horizon (oxidized surface) distinguishes the FGS from the fine-grained deposits referable to MDS." - In the corresponding figure 4 you have described the boundary between FGS and MDS as an "unconformity". Please use the same terms in the text and the figures. Using different terms might confuse the reader.

Answer: we modified the text as follows: In the S5 borehole, a pedogenetic horizon (oxidized surface), corresponding to a probable stratigraphic unconformity, distinguishes the FGS from the fine-grained deposits referable to MDS (Nocentini, 2016).

page 7 line 5: use "deepest part" instead of "deepest portion".

Answer: done (three times in the manuscript).

page 7 line 30: "Its basal boundary is highly irregular, and it was recognized down to 80 ms." - In the section 'Seismic data acquisition' you have written that the first 50 m were not properly imaged by the reflection seismic due to the resolution limits in the near-surface. Are you sure that you can get a reliable interpretation for facies BC that is located in the uppermost part of your seismic profile?

Answer: we are confident that the upper part of the Corso section is characterized by BC seismic facies also by considering the tomography (new Fig. 11), the borehole stratigraphy and the fine scale geological setting synthetized from Nocentini et al., 2017 (new Fig. 3).

page 8 line 3: "The calculated Vp for the tomography are different from that used for the reflection profile." - What exactly is the difference between both velocity fields? Give numbers or show an image in which both velocity fields are compared.

Answer: this part was completely re-written. Now, the new table 4 reports the Vp values obtained by our velocity analysis. Concerning the comment: "The calculated Vp for the tomography are different from that used for the reflection profile", we show the calculated Vp value in the new Fig. 12 in which velocity fields are compared. We

modified the text as follows: "The refraction velocity field (Fig. 11) interested the first 100 depth. In particular, in the first 20 m depth very heterogeneous formations are present and consequently there is a strong scattering; from 20 up to 80 m about the Vp increment slowly from 1600-2000. The first clear impedance contrast is detected at about 80 m depth where Vp is 2600 m/s. In reflection processing, from 0 up to 80 m of depth, the velocity analysis not detect Vp variation. This is due to strong scattering and low Vp gradient. The first clear impedance contrast is detected at about 80 m depth where in velocity analysis the semblance function (Fig. 12) shows high coherence. The calculated Vp are showed in table 4 and it are compatible with Vp determined by Improta et al, 2012."

page 8 line 15 and line 24: "ubiquitarian" - I guess you mean ubiquitous. The meaning of ubiquitarian is: relating to or believing in the doctrine that Christ is present everywhere at all times. I guess that is not what you wanted to say.

Answer: done (two times in the manuscript).

page 9 line 8 to 9: "...the evolution of infilling deposits depends on the subsidence of the basin, which is mainly controlled by the geometry of the fault systems affecting the basin." - What is with subsidence resulting from the accumulation of large volumes of sediments? In many basins we have an interaction between fault-related and loading related subsidence.

Answer: we agree with your comment and added the loading related subsidence. We modified with followed: "In a tectonically active intermontane continental basin, the evolution of infilling deposits depends on the subsidence of the basin, which is mainly controlled by the geometry of the fault systems and also by the sediment loading."

page 10 line 4: "...has been drown..." - I guess you mean "has been drawn".

Answer: done.

page 10 line 17: "...the sedimentological characteristics of CMA point to huge events

of..." - a short repetition of the most important sedimentological characteristics of CMA would be helpful at this point.

Answer: we modified the text as follows: "As stated before, the sedimentological characteristics of CMA, composed by massive and chaotic calcareous breccias, point to huge events of detrital deposition through debris flow and rock avalanche with debris produced mainly by the erosion of the northern margin of ASB (Gran Sasso chain), possibly during a cold late Middle Pleistocene event (Cosentino et al., 2017)."

page 10 line 21 to 23: see my comment on seismic tomography and structural interpretation for page 5 line 10.

Answer: see the modified text on comment to page 5 line 10.

General comments and or questions to the paragraph 'Discussion': The discussion of the tectonic features and the subsurface model is good, but what is completely missing so far is the critical discussion of the reflection and refraction seismic methods. Most of your results, except for the borehole stratigraphy, are based on geophysics. As a consequence, you should discuss problems and disadvantages/advantages of both methods regarding data acquisition and processing. For example, I think discussing the resolution limits of your data would be very helpful. You should ask yourself 'What could have been done better and what other geophysical investigations would I carry out in the future, in the case of a subsequent project.

Answer: we discussed of the problems of reflection and refraction methods regarding data acquisition, processing and of the resolution limits. In particular, we estimated the vertical and horizontal resolution for the reflection method and the ray density for the seismic tomography (see the modified text on comment to page 5 line 10 and comment to page 8 line 3). In a future project, we retain that could be very useful the investigation of L'Aquila downtown with SH-wave or other similar techniques as suggested by referee 2 especially for the shallow subsurface as now is working in progress for the mapping of red soils colluvium and epikarst and anthropic covers,

which are poor from the geotechnical point of view. The same methodology could be helpful to map and calculate the offset in Holocene deposits of active faults, such as the PDF.

Other questions you could think of are 'Can you compare your results with other intermontane basins in seismically-active regions?' and 'With your results is it possible to better estimate the future seismic hazard for downtown L'Aquila? For example, can you define particular risk areas, where damage to buildings and infrastructure would be higher than in other areas due to the local near-surface geology derived from your data.

Answer: for the first question: we compare our results with other central Italy intermontane basins as written in this paragraph: "The Meso-Cenozoic bedrock is located maximum at 600 m b.g.l., and though it is the deepest value for the bedrock depth in ASB, it is in accordance with Meso-Cenozoic bedrock depth of other intermontane basins of central Italy as the Fucino Basin (Cavinato et al., 2002) and the Paganica-San Nicandro-Castelnuovo Basin (Civico et al., 2017)." For the second question: the 3D model of the bedrock top (Fig. 11) will allow to carried out detailed 1D and 2D numerical simulation with the aim to obtain more reliable seismic amplification factor of L'Aquila downtown.

Technical corrections Figure 4: The marked "lignite level" (black line) in the stratigraphy plot is hardly visible. Maybe using a different colour, e.g. red, would be better.

Answer: done.

Figure 5: For a better correlation of seismic and borehole results it would be nice if you could draw the location of the nearby boreholes into the seismic profile. This will help to better verify the seismic interpretation.

Answer: done.

Figure 5: Why have you abbreviated the seismic facies twice? In the text and in table

2 where you are describing the seismic facies in detail you use BC, Ls, L, R and S as abbreviations for your seismic facies analysis. In the figure and the figure caption you use the following legend "1= seismic facies BC; 2= channelized bodies in seismic facies BC; 3= seismic facies L; 4=seismic facies Ls; 5= seismic facies R; 6= seismic facies S". I understand why you had to find new abbreviations for e.g. the channel bodies in seismic facies BC, but I do not understand why you had to rename the facies classes themselves. This would be very confusing for the reader. I would rewrite the legend in the figure caption and the corresponding part of figure 5c like this: BC = fan deposits and slope breccias; 1 = channelized bodies in seismic facies BC; L = alluvial plain deposits; 4 = channelized deposits; R = fan deposits and slope breccias; S = meso-cenozoic bedrock; 2= fault; 3= channelized bodies; 4= unconformity; 5= top of Meso-Cenozoic bedrock. This way you do not have two different abbreviations for the same facies. Answer: done.

Figure 5: In the text you have written that you used a Kirchhoff time migration, but in the figure caption you have written that figure 5a shows the "2D depth-migrated reflection of the Corso section". Have you carried out time-migration or depth-migration? You have also written "common deep point" but it must be 'common depth point' and instead of "two-way time" you should use 'two-way traveltime'.

Answer: we carried out a Kirchhoff Time-migration. We corrected the caption of the new Fig. 10: "Figure 10: A) 2D time-migration section of the Corso section (horizontal scale= CDP, vertical scale= two-way traveltime);".

Figure 7: When drawing faults into a cross section it is necessary to draw arrows indicating the fault movement.

Answer: done.

Overall, a very nice work, congratulations!

Answer: many thanks for your helpful comments. We upload the new figures 5, 6, 7, 8,

12.

[Figure]

**Fig. 1.** Figure 5: Velocity analysis (on the left) of a CDP supergather (on the right) by semblance function. The gather is analyzed over time windows for the values of semblance according to a range of stacki

**Fig. 2.** Figure 6: Example of two raw shot gathers (a) and the same after processing (b). Shot generated noise refers to multiple reverberations and groundroll.

[Figure]

**Fig. 3.** Figure 7: Comparison between the project sweep and the spectral content of all the tracks recorded along the profile. The strong signal attenuation after 60 Hz is clearly evident.

[Figure]

**Fig. 4.** Figure 8: Ray density (RD) model obtained by derivative weight sum (Kissling, 1988). Lateral velocity variation influence the RD distribution and the final tomographic model.

[Figure]

**Fig. 5.** Figure 12: Vp from velocity analysis (red line) and tomography (black line) obtained from the Vp average along the profile respect to the depth. It is clear a strong impendence contrast at about 100 m